# Prevalence of fast food consumption and associated factors among secondary school adolescents in Jigjiga Town Somali Region Eastern Ethiopia

Ahmed Mohammed Ibrahim[1,2,3]*, Nasra Ali Hussein[1], Mohyadin Abdullahi Ahmed[1], Girma Tadesse Wadajo[1], Mohamed Omar Osman[1], Ramadan Budul Yusuf[1], Abdilahi Ibrahim Muse[1], Seid Muhumed Abdilaahi[4], Raymond Place[5], Jakob Zinsstag[2,3]

1 Department of Public Health, College of Medicine and Health Science, Jigjiga University, Jigjiga, Ethiopia, 2 Swiss Tropical and Public Health Institute, Allschwil, Switzerland, 3 University of Basel, Basel, Switzerland, 4 Department of Pediatrics and Child Health Nursing, Institute of Health Science, Jigjiga University, Jigjiga, Ethiopia, 5 School of Agricultural, Forest and Food Sciences, Bern University of Applied Sciences (BFH-HAFL), Zollikofen, Switzerland

* ahmey114baba@gmail.com, ahmedmohammed@jju.edu.et, ahmed.ibrahim@swisstph.ch, ahmed.ibrahim@unibas.ch

## Abstract

### Background

Fast food consumption is a growing public health concern and creates a burden of malnutrition and non-communicable diseases, especially among adolescents in low- and middle-income countries. Adolescents are particularly vulnerable due to increasing autonomy and exposure to unhealthy fast food environments. Adolescents are particularly vulnerable due to increasing autonomy and exposure to unhealthy food environments.

### Objective

To assess the prevalence of fast food consumption and associated factors among secondary school adolescents in Jigjiga town, Somali region, Eastern Ethiopia.

### Methods

An institution-based cross-sectional study was conducted. A total of 419 participants were selected using a systematic random sampling technique from six randomly selected secondary schools (2 public and 4 private schools). Data were collected using a structured questionnaire. Data entry was done using Epi-Data 3.1 and then was analyzed with SPSS version 23.0. Descriptive statistics were calculated, and logistic regression analyses were done to identify factors associated with fast food consumption. Variables with p-values <0.25 in bivariate analysis were included in the

**Data availability statement:** All data are in the manuscript and/or Supporting Information files.

**Funding:** The author(s) received no specific funding for this work.

**Competing interests:** The authors have no conflict of interest.

multivariable model. Adjusted odds ratios (AORs) with 95% CIs were reported, and finally statistical significance was declared at p-value < 0.05. Model fit was calculated using Hosmer and Lemeshow's test, and multicollinearity was checked using the Variance Inflation Factor (VIF).

## Results

The prevalence of fast food consumption among adolescents was 49.2**%** (95% CI: 44.2, 54.1). Significant factors associated with consumption of fast food are being younger age (AOR = 2.02; 95% CI: 1.30, 3.16), being male (AOR = 1.94; 95% CI: 1.25, 3.01), living with individuals other than parents (AOR = 2.58; 95% CI: 1.51, 4.39), availability of fast food at home (AOR = 2.03 95% CI: 1.31, 3.14), having pocket money ≥ 500 ETB (AOR = 1.97; 95% CI: 1.12, 3.46), and being food insecure (AOR = 2.08; 95% CI: 1.32, 3.28).

## Conclusion

Fast food consumption among adolescents in the study area is high, with higher consumption. Younger age, male gender, living with individuals other than parents, availability of fast food at home, higher pocket money, and household food insecurity were significantly associated with fast food consumption. Community-based and school interventions are needed to promote healthy food-consuming habits and reduce the health risks associated with frequent fast food intake in the study area.

## Introduction

Fast food refers to readily available, highly processed food items from quick-service restaurants or vendors, often high in calories, saturated fats, sugars, and sodium, and low in nutritional value [1]. Fast foods often contain high levels of unhealthy fats, sugars, and sodium, which can contribute to various health issues when consumed in excess. Regular consumption of fast food is associated with an increased risk of obesity, cardiovascular diseases, type 2 diabetes, and other metabolic disorders. These foods are typically low in essential nutrients, such as vitamins, minerals, and fiber, leading to potential nutritional deficiencies [2]. Moreover, the frequent consumption of fast foods can displace more nutritious meals, negatively impacting overall dietary quality. The convenience and affordability of fast food often make it a popular choice, particularly in urban areas, but this convenience comes at the cost of long-term health consequences [3].

In fast food restaurants, food is prepared quickly and delivered to customers' homes once being packed, sold, and stored [4]. Growing lifestyle changes and increased processed food consumption are associated with the growth of supermarkets and other modern retail establishments [5–7].

Fast food consumption contributes to rising levels of adolescent malnutrition, especially in LMICs, by displacing healthier meals and promoting unhealthy dietary

patterns [8]. Ethiopian and global fast-food consumption is currently rising as a result of the growing number of fast-food restaurants. Little is known about the fast-food composition and associated factors in Jigjiga town; therefore, the aim of this proposal is to assess fast-food composition and associated factors among secondary school adolescents in Jigjiga town.

Young adolescents around the world, particularly those in low- and middle-income countries (LMICs), are going through a nutritional transition characterized by a sharp change in their consuming habits from the traditional diets of their respective nations to Westernized diets [6]

Several studies have shown that teens who consume a lot of high-fat snacks, sweetened beverages, and a high proportion of saturated fats may be more likely to become obese and overweight [9].

Furthermore, given that young adolescents in LMICs frequently suffer from several burdens of malnutrition [10]. Fast food consumption has been linked to increasing rates of obesity, diabetes, cardiovascular diseases, and metabolic syndrome among adolescents [11,12].

A study conducted in the United States of America on adolescents between the ages of 2 and 19 found that, on average, fast food accounted for 14% of the calories ingested by these individuals each day [13]. According to the other study conducted in Saudi Arabia, 79.1% of teenagers eat fast food at least once a week [14]. Chronic disorders such as hypertension, cardiovascular disease, and type 2 diabetes mellitus are inevitably going to emerge after consuming unhealthy fast food [15]. Fast food has been linked to kid obesity in many countries despite the fact that the primary causes of the issue are thought to be numerous and complicated. This is because of the meal's increased availability, high-energy content, and generous portion sizes [16].

A person's food-consuming habits and nutritional status can be influenced during adolescence, which may have a long-term effect on their health in the future [17–19]. Despite the growing prevalence of fast food consumption in Ethiopia, particularly in urban areas, there is limited evidence on its magnitude and associated factors among adolescents in Jigjiga. This study seeks to fill that gap by assessing the prevalence and determinants of fast food consumption among secondary school adolescents in Jigjiga Town.

## Method and materials

### Study area and period

Jigjiga is the capital city of the Somali regional state of Ethiopia. It is located 630 km from the capital city of the country, Addis Ababa. Jigjiga is a city council administration, and it is one of the six city council administrations in ESRS. According to the structure of the country, Jigjiga is a city council, which has its own city administration, and the city has 30 kebeles. 20 of these kebeles are urban, and the remaining 10 kebeles are rural. The total population of the city is projected to be 182,922,474 (CSA, 2024). According to the Jigjiga City Council Administration Educational Bureau 2024 plan office, there are 25 secondary schools, where nine of them are public schools and 16 of the remaining secondary schools are privately owned. There are 132,650 secondary school students in total in the city. Out of which 53,023 students are in the public schools and 6,459 students are from private schools. Moreover, the study period was conducted starting from April 1 to April 30, 2024.

### Study design

An institution-based cross-sectional study was conducted.

### Source and study population

**Source of population.** The source population for this study was all secondary school adolescents in Jigjiga town.

**Study population.** The study population for this study was all randomly selected secondary school adolescents of grades 9–12 in Jigjiga town during the study period.

### Inclusion and exclusion criteria

The study included all secondary school-attending adolescents in randomly selected schools in Jigjiga town. All secondary adolescents who are seriously ill, cannot respond, are absent, or refuse to participate in the study were excluded from the study.

### Sample size determination

The required sample size of the study participants was determined by using the single population proportion formula.

$$n = \frac{\left(Z\frac{\alpha}{2}\right)2 \times P(1-P)}{d^2}$$

With following assumptions;

 n – The minimum sample size required

 P – Estimated proportion of fast-food consumption among secondary school adolescents.

 d – Margin of error 5%(0.05)

 Za/2- Standard normal value at (1-ɑ) 95% confidence level

 Prevalence (P) of magnitude fast-food consumption was taken from a study conducted in Harar town which was 45.8% (37).

$$\text{Hence, N} = \frac{[Z\alpha2]^2 p(1-p)}{d^2} = \frac{(1.96)^2\ (0.458)\ (1-0.458)}{(0.05)^2} = 381.4 = 381$$

Considering non-response rate of 10% [which equals to 38.1] 381+38.1 = **419**.

### Sampling technique and procedure

Jigjiga city administration has 25 secondary schools, with 9 of them being public schools and 16 being privately owned. To obtain a representative sample, 2 public and 4 private secondary schools were randomly selected for this study. For each selected school, the total number of secondary school adolescents was obtained from the school records. This information was important for allocating the sample proportionally to each school based on the number of students enrolled. The proportional allocation ensured that each school contributed to the final sample in proportion to its size. After the schools were allocated, a systematic random sampling technique was employed to select the participants within each school (Fig 1). A sampling interval was determined by dividing the total number of students in each school by the desired sample size, and the Kth values were found to be 28. Then a number between 1 and 28 values was taken as a random starting point, and finally 5 were chosen. Then every fifth student was included in the sample until the required sample size was reached.

### Data collection method

Data was collected using an interviewer-administered and structured questionnaire. The data collection tools were adapted from various literatures. Data collection tools were initially created in English and then translated into Somali and Amharic and back into English by fluent speakers of the three languages to check consistency. The questionnaire consists of three sections: socio-demographic characteristics of the participants, influencing factors for fast food consumption, and nutrition-related factors

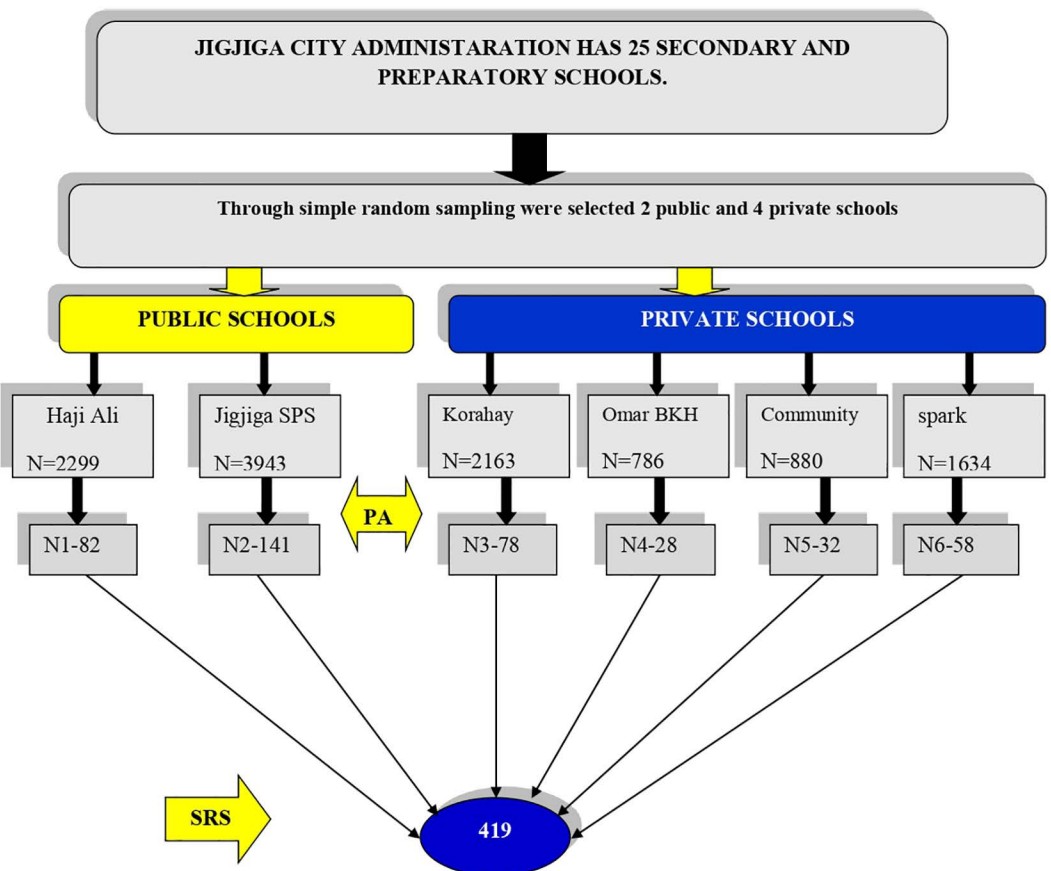

**Fig 1. Schematic representation of sampling procedure for the study on fast food consumption and associated factors among secondary school adolescents in Jigjiga town, eastern Ethiopia, 2024.**

**Study variables.** The dependent variables of this study is Fast-fast consumption where Independent variables were Socio-demographic characteristics (age, sex, type of school, father's education, father's occupation, residential area, and living arrangements), Nutritional-related factors (frequency of fruit and vegetable consumption and food security) **a**nd Other related factors (distance of food outlet, pocket money, time, place, reason, availability, advertisement, and household income).

## Operational definition

**Adolescents.** According to WHO definitions, adolescents are age groups between 10 and 19 years old [20].

**Fast food.** The fourth group of the NOVA food classification of fast foods includes teramisu, cake, cookies, biscuits, chocolate, soft drinks (Coca-Cola, Fanta, Pepsi), burgers, pizza, canned foods, meat products, Donatus, Brazin, juice, Baklawa, Alawa, chips, ice cream, and tastes [21].

**Fast food consumption.** The primary outcome variable was fast food consumption, defined as consuming one or more fast food items on three or more days in the past week [22].

**Food-secure.** According to the HFIAS score Adolescents with a food security score of 0–1 were considered food secure, while those with HFIAS scores of 2 and above were considered food insecure [23].

**Knowledge about the fast food consumption.** The level of knowledge regarding the fast food consumption was assessed using a set of 10 knowledge-related questions. The correct answer gets a score of 1, and the wrong answers get a score of 0. The total score obtained was then divided into two categories. Good and poor knowledge.

## Data quality control

The quality of data was assured by proper designing and pre-testing (5%) of the questionnaires in those adolescent'students who were not participating in the study with similar socio-demographic characteristics in order to ensure its validity. Training was given for both data collectors and supervisors by the principal investigator for one day. The training included discussion on the objectives of the study and on the contents of the questionnaire one by one, on procedures, on data collection techniques, and on the issues of the confidentiality of the responses. The collected data was checked by supervisors on a daily basis for any incompleteness or inconsistency, and possible corrections were made each day based on the identified gaps accordingly. Inconsistencies and possible corrections were made each day based on the identified gaps accordingly.

## Data analysis and processing

First, data was checked for completeness and consistency before being entered into the computer. Then, after entering Epi-Data version 3.1 software, it was coded and exported to SPSS statistical software version 23.0 for analysis. Descriptive statistics such as mean, median, frequency, and percentage were used and presented using charts and tables. Bivariate binary logistic regression analysis was done, and all explanatory variables that were candidates in the bivariate model with a p-value less than 0.25 were included in the multivariable logistic regression to avoid excluding potentially important variables that may indicate significance when adjusted for other factors [24]. Before running the multivariable logistic regression, multicollinearity was assessed among the candidate independent variables using the variance inflation factor (VIF) and tolerance. Model fit was assessed using the Hosmer and Lemeshow goodness-of-fit test, which yielded a p-value of 0.365, indicating a good model fit. Additionally, discriminative ability was evaluated using the Receiver Operating Characteristic (ROC) curve, which produced an Area under the Curve (AUC) of 0.74, suggesting an acceptable level of model discrimination. A Nagelkerke pseudo $R^2$ value of 0.31 was also reported, indicating that approximately 31% of the variability in fast food consumption was explained by the mode. The strength of the statistical association was measured by odds ratio and 95% confidence intervals, and statistical significance was declared at a P-value of less than 0.05.

## Ethical consideration

Ethical approval for this study was obtained from the Jigjiga University Research Ethics Review Committee (JJURERB/039/2024). The official permission letter was also received from the Jigjiga City Administration Education Office. Informed, voluntary written and signed consent was obtained from each parent of the adolescent, and an adolescent assent form was obtained for each study participant after explaining the purpose and benefits of the study. In order to maintain study participant confidentiality, no identifying information was included in the questionnaires, and participants were made aware that their data would only be utilized for research.

## Results

### Socio-demographic characteristics of the respondents

A total of 419 adolescent students were selected for the study, of which 413 participated, corresponding to a response rate of 98.5%. Participants had a mean age of 16.67 years (± SD 1.67) and ranged from 14 to 19 years. More than 223 (54.0%) of the adolescents were between 17 and 19 years old. Approximately 230 (55.7%) of participants were male. Over half (219, 53.0%) of participants were enrolled in public schools. More than three-quarters (318, 77.0%) of the

young people lived with their parents. Around 183 (44.3%) of the fathers of the respondents had a secondary education or above. About 248 (60.0%) of the young people's fathers were employed. Two-thirds (277, 67.1%) of the adolescent students lived in urban areas. Finally, 286 (69.2%) of the participants had a family income of 4,000 birr or more per month (Table 1).

### Influencing factors of fast-food consumption

According to the findings, just over half (228, 55.2%) of the adolescents do not have fast food available at home. However, the most common time for fast food consumption is dinner. 135 (32.7%). Around 115 (27.8%) of the respondents' most common place of fast food consumption is street food stalls. The main reason for fast food consumption is taste preference (24.5%), followed by convenience (23.5%). The most common source of information about fast food is television (35.8%). About 156 (37.8%) of the adolescents spend between 300–400 Birr per week on fast food. Finally, around 135 (32.7%) of the adolescents have access to fast food outlets from both home and school (Table 2).

### Nutrition-related factors

Nearly half (188, 45.5%) of the respondents consume fruits and vegetables 1–2 times a week. Based on a measurement of food security, around 215 (52.1%) of the adolescents in the study were living in food-secure households, while the rest were living in food-insecure households (Table 3).

### Knowledge of harmful health effects of fast-food consumption

Regarding respondents' awareness of health risks associated with fast food consumption, nearly half (197, 47.7%) of the adolescents had good knowledge (Fig 2). The remaining adolescents, which is 52.3%, had poor knowledge.

**Table 1. Socio-demographic characteristics among adolescents' student in Jigjiga town, Eastern Ethiopia, 2024.**

| Variables | Category | Frequency | Percent |
|---|---|---|---|
| Age of adolescents | 17–19 | 223 | 54.0 |
| | 14–16 | 190 | 46.0 |
| Sex of adolescents | Female | 183 | 44.3 |
| | Male | 230 | 55.7 |
| Type of school | Public | 219 | 53.0 |
| | Private | 194 | 47.0 |
| Who are you living with | Parents | 318 | 77.0 |
| | Relatives | 48 | 11.6 |
| | Alone | 47 | 11.4 |
| Fathers education | Unable to read and write | 128 | 31.0 |
| | Primary school | 102 | 24.7 |
| | Secondary and above | 183 | 44.3 |
| Fathers occupation | Employed | 248 | 60.0 |
| | Unemployed | 165 | 40.0 |
| Place of residency | Urban | 277 | 67.1 |
| | Ruler | 136 | 32.9 |
| Average monthly income | ≥ 4000 Birr | 286 | 69.2 |
| | < 4000 Birr | 127 | 30.8 |

**Table 2. Factors associated with fast-food consumption among secondary school adolescents in Jigjiga town, Eastern Ethiopia, 2024.**

| Variables | Category | Frequency | Percent |
|---|---|---|---|
| Availability of fast food at home | Yes | 185 | 44.8 |
| | No | 228 | 55.2 |
| Time of consumption | Breakfast | 135 | 32.7 |
| | Lunch | 108 | 26.2 |
| | Dinner | 87 | 21.1 |
| | Snacks | 66 | 16.0 |
| | No specific time | 17 | 4.1 |
| Place of consumption | At home | 103 | 24.9 |
| | At school | 101 | 24.5 |
| | At restaurants | 94 | 22.8 |
| | Street food stall | 115 | 27.8 |
| Reason for consumption | Taste preference | 101 | 24.5 |
| | Convenience | 97 | 23.5 |
| | Availability of fast food outlets near school/home | 88 | 21.3 |
| | Peer influence | 81 | 19.6 |
| | Having many variety | 46 | 11.1 |
| Sources of information | Television | 148 | 35.8 |
| | Outdoor billboards in public places | 125 | 30.3 |
| | Internet and social network | 92 | 22.3 |
| | Newspapers, Magazine | 35 | 8.5 |
| | FM radio | 13 | 3.1 |
| Money spend on fast food per week | 100–200 Birr | 115 | 27.8 |
| | 300–400 Birr | 156 | 37.8 |
| | > 500 Birr | 142 | 34.4 |
| Fast food retailers in 10 min | Yes, I have from home | 103 | 24.9 |
| | Yes, I have from school | 113 | 27.4 |
| | Yes, I have from both | 135 | 32.7 |
| | Not, from both | 62 | 15.0 |

**Table 3. Nutritional related factors among secondary school adolescents in Jigjiga town, Eastern, Ethiopia 2024.**

| Variables | Category | Frequency | Percent |
|---|---|---|---|
| Fruit and vegetable consumption per week | 1–2 Times | 188 | 45.5 |
| | 3–4 Times | 130 | 31.5 |
| | 5 and more | 95 | 23.0 |
| Food security status | Food secure | 215 | 52.1 |
| | Food insecure | 198 | 47.9 |

## Prevalence of fast food consumption

In this study, the overall prevalence of fast-food consumption among secondary school adolescents was 49.2% (95% CI: 44.2, 54.1) (Fig 3). This indicates that 203 out of the total 413 participants (49.2%) consumed fast food for three or more days in the previous week. Conversely, 210 participants (50.8%) consumed fast food for less than 3 days in the previous week.

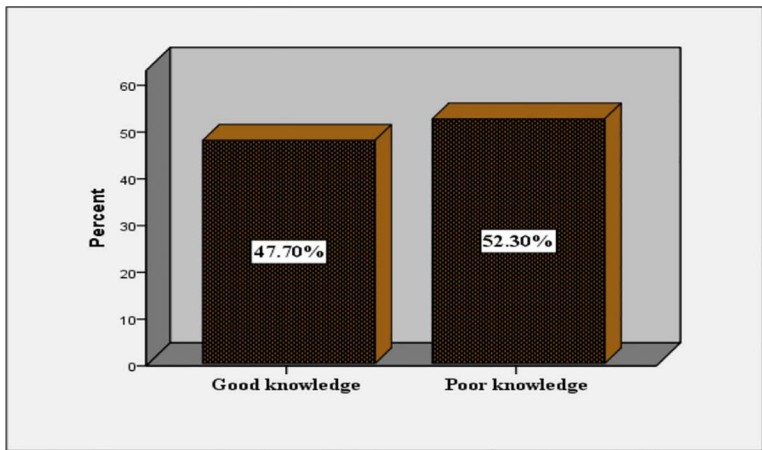

**Fig 2. Adolescents' knowledge about health risks of fast food consumption.**

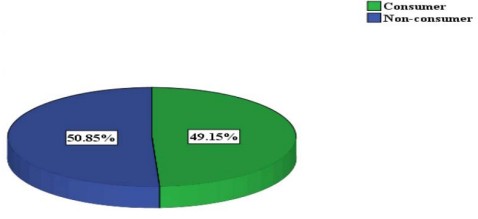

**Fig 3. Prevalence of fast food consumption among secondary school adolescents.**

## Factors associated with fast food consumption

In the analysis of bivariate logistic regression, several variables become candidates in bivariate mode with a p-value of less than 0.25. These variables included age, sex, living arrangements, father's education, household income, pocket money, availability, knowledge, food security, fruit and vegetable consumption, and distance from fast food retailers. Among the associated factors, high weekly pocket money, food insecurity, and fast food availability at home were significantly associated with fast food consumption. Regarding the age group, adolescents aged 14–16 (AOR = 2.03; 95% CI: 1.30, 3.16) were highly using the fast food. The odds of fast food consumption were 2 times higher among adolescents aged 14–16 years as compared to those aged 17–19.

Another factor that was found to be significantly associated with fast-food consumption was sex (AOR = 1.94; 95%; CI: 1.25, 3.01). Female adolescents had a 1.9 times higher likelihood of consuming the fast food compared to their male counterparts. Another important factor associated with fast-food consumption was weekly spending pocket money (AOR = 1.97; 95% CI: 1.12, 3.46). Participants with high pocket money were two times more likely to consume fast food compared to participants with low pocket money.

The higher likelihood of fast-food consumption was demonstrated by the adolescents who were living with others as compared to those who lived with parents (AOR = 2.58; 95% CI: 1.51, 4.39). The odds of fast-food consumption were 2.5 times higher among adolescents who were living with others (relatives and alone), as compared to those living with parents.

The food security status of the household was significantly associated with fast-food consumption (AOR = 2.085; 95% CI: 1.32, 3.28). The possibility of consuming fast food was 2 times higher among adolescents who experienced food

insecurity compared to those who did not face food insecurity. Availability of fast food at home was also significantly associated with fast-food consumption (AOR = 2.03; 95% CI: 1.31, 3.14). Those adolescents who live in homes with the availability of fast food were two times more likely to eat fast food compared to those who don't have easy access to fast food at home (Table 4).

## Discussion

A total of 413 participants from the two schools were included in this cross-sectional study, which was conducted in Jigjiga town, Somali Region, Eastern Ethiopia, and aimed to identify the prevalence and associated factors of fast food consumption among secondary school adolescents. The study found that the prevalence of fast food consumption among adolescents was 49.2% (95% CI: 44.2, 54.1). The result of the present study is lower than the study conducted in India (80) [25], in Uganda (90%), and in low- and middle-income countries (55.2%) [26]. The difference could be due to variation in study setting, time, data collection tools and technique, sample size among different studies, and difference among factors

**Table 4. Bivariate and multivariate binary logistic regression analyses of factors associated with fast food consumptions among secondary school adolescent in Jigjiga town, Somali region, Eastern Ethiopia, 2024.**

| Variables | Categories | Fast-food consumption | | COR (95%CI) | AOR (95%CI) | p-value |
|---|---|---|---|---|---|---|
| | | Yes | No | | | |
| Age of the respondents | 17-19 | 94 (42.2%) | 129 (57.8%) | Ref | Ref | |
| | 14-16 | 109 (57.4%) | 81 (42.6%) | 1.84 (1.24,2.73) | 2.02 (1.30, 3.16) | 0.002* |
| Sex of the respondents | Female | 71 (38.8%) | 112 (61.2%) | Ref | Ref | |
| | Male | 132 (57.4%) | 98 (42.6%) | 2.12 (1.43,3.15) | 1.94 (1.25, 3.01) | 0.003* |
| Living with | Parents | 140 (44.0%) | 178 (56.0%) | Ref | Ref | |
| | With others | 63 (66.3%) | 32 (33.7%) | 2.50 (1.54, 4.04) | 2.58 (1.51, 4.39) | 0.000* |
| Income | ≥ 4000 birr | 132 (46.2%) | 154 (53.8%) | Ref | Ref | |
| | < 4000 birr | 71 (55.9%) | 56 (44.1%) | 1.47 (0.97, 2.25) | 1.34 (.830, 2.19) | 0. 227 |
| Fathers education | Uneducated | 69 (53.9%) | 59 (46.1%) | Ref | Ref | |
| | Primary | 50 (49.0%) | 52 (51.0%) | .82 (.488, 1.38) | 0.92 (.513, 1.66) | 0.794 |
| | Secondary+ | 84 (45.9%) | 99 (54.1%) | .72 (.461, 1.14) | 0.73 (.436, 1.24) | 0.256 |
| Availability of fast food at home | No | 90 (39.5%) | 138 (60.5%) | Ref | Ref | |
| | Yes | 113 (61.1%) | 72 (38.9%) | 2.40 (1.61, 3.58) | 2.03 (1.31, 3.14) | 0.001* |
| Pocket money | 100-200 Birr | 45 (39.1%) | 70 (60.9%) | Ref | Ref | |
| | 300-400 Birr | 71 (45.5%) | 85 (54.5%) | 1.29 (.796, 2.12) | 1.18 (.689, 2.05) | 0.536 |
| | ≥ 500 Birr | 87 (61.3%) | 55 (38.7%) | 2.46 (1.48, 4.07) | 1.97 (1.12, 3.46) | 0.018 |
| Knowledge | Good knowledge | 88 (44.7%) | 109 (55.3%) | Ref | Ref | |
| | Poor knowledge | 115 (53.2%) | 101 (46.8%) | 1.41 (.957, 2.07) | 1.50 (.970, 2.33) | 0.068 |
| Food security | Secure | 87 (40.5%) | 128 (59.5%) | Ref | Ref | |
| | Insecure | 116 (58.6%) | 82 (41.4%) | 2.08 (1.40,3.08) | 2.08 (1.32, 3.28) | 0.001* |
| Fruit &vegetable consumption | 1-2 Times | 102 (54.3%) | 86 (45.7%) | Ref | Ref | |
| | 3-4 Times | 55 (42.3%) | 75 (57.7%) | 0.61 (.394,.971) | 0.66 (.398, 1.09) | 0.110 |
| | 5 and more | 46 (48.4%) | 49 (51.6%) | 0.79 (.483, 1.29) | 0.81 (.46, 1.41) | 0.462 |
| Fast food retailers in 10 min | Yes I have from home | 44 (42.7%) | 59 (57.3%) | Ref | Ref | |
| | Yes I have from school | 50 (44.2%) | 63 (55.8%) | 1.06 (.621, 1.82) | .95 (.524, 1.729) | 0.870 |
| | Yes, I have from both | 76 (56.3%) | 59 (43.7%) | 1.72 (1.02, 2.89) | 1.50 (.84, 2.68) | 0.168 |
| | Not, from both | 33 (53.2%) | 29 (46.8%) | 1.52 (.810, 2.87) | 1.20 (.591, 2.46) | 0.605 |

Note: *Statistically significant

responsible for the consumption of fast food. Besides, the variations might be due to differences in the socio-cultural and economic conditions of the respondents.

On the other hand, the prevalence of fast-food consumption in this study was significantly higher than in studies conducted in the USA (16.7%) [27], in Saudi Arabia (7.5%) [28], and in South Africa (21%) [29]. The discrepancy could be due to the difference in methodology of these studies, educational backgrounds of the participants, and socioeconomic differences between study participants. Additionally, cultural norms and preferences related to fast food consumption may vary based on socio-economic status. The study identified several factors associated with fast-food consumption, including age, sex, living arrangement, pocket money, availability of fast food at home, and food insecurity.

Regarding the age group, adolescents aged 14–16 were highly using the fast food. The odds of consuming fast food were 2 times higher among adolescents aged 14–16 years as compared to those aged 17–19. The result of this study was similar with studies done in Malaysia [30]. Possible justification for similarities could be that younger adolescents may be more influenced by peer pressure and social norms surrounding fast food consumption or have less knowledge about healthy eating practices and the negative impacts of consuming fast food regularly.

Another factor that was found to be significantly associated with fast-food consumption was sex. Male adolescents had a 1.9 times higher likelihood of using fast food compared to their female counterparts. The result of this study is contradicted by a study conducted in Malaysia [30] and another study conducted in 44 low- and middle-income countries, which reported that female adolescents were more likely to consume fast food than male adolescents [31]. A possible explanation for this inconsistency might be due to the sample size of these studies, demographic differences of the study sites, and cultural influences that individual preferences and preferences regarding fast food consumption can vary greatly among different groups of people. On the other hand, the observed gender difference, where male adolescents were more likely to consume fast food, may reflect local cultural norms in Somali society, where boys often enjoy greater mobility and spending autonomy than girls, and this freedom may increase their exposure to street food and fast-food outlets.

The higher likelihood of fast-food consumption was shown by the adolescents who were living with others as compared to those who lived with parents. The odds of fast-food consumption were 2.5 times higher among adolescents who are living with others (relatives and alone) as compared to those living with parents. This result was inconsistent with another study conducted in Pokhara Valley, Nepal, which showed that adolescents living with parents were more likely to consume fast food compared to those who lived with others [32]. The variation in fast-food consumption between the two studies could be due to differences in parental supervision or cultural norms and financial independence of the adolescents.

Another significant factor associated with fast-food consumption in the current study was weekly spending money. Participants with high pocket money were 1.9 times more likely to eat fast food compared to participants with less pocket money. The finding of this study was comparable with research carried out in Australia [33], in the Kathmandu District of Nepal [34], and in Jakarta, Indonesia [35]. Adolescents with more pocket money may have greater autonomy in deciding what and where to eat, leading them to opt for convenient and readily available fast food options [36]. A conceivable reason could be that having more income to use allows individuals to make independent choices about their food consumption without relying on parental approval or financial constraints.

The likelihood of consuming fast food was 2 times higher among adolescents who experienced food insecurity compared to people who did not face food insecurity. This is in line with a similar study conducted in Harar town, Ethiopia [37], and another study conducted in sixty-eight countries [38]. The possible justification could be that food insecurity may result in limited access to affordable and nutritious food options, making fast food a convenient and inexpensive choice for many individuals facing economic challenges, a pattern consistent with findings in other LMICs [13,39].

Finally, availability of fast food at home was also significantly associated with fast-food consumption. Those adolescents who live in homes with the availability of fast food were two times more likely to consume fast food compared to those who don't have easy access to fast food at home. The result of this study was comparable with studies conducted in Harar town, Ethiopia [37], and in Nepal [32]. The possible explanation for this similarity is based on the fact that when

fast food is readily available in the home, it becomes a more convenient and accessible food option for adolescents. They don't have to go out of their way to obtain fast food, making it easier to choose these options. Additionally, growing up with fast food as a regular presence in the home can normalize its consumption for adolescents.

## Limitations of the study

One of the limitations of this study is the nature of the study; the cross-sectional design limits the ability to infer causal relationships between the identified factors and fast food consumption. others are Self-reported data may introduce recall bias and social desirability biases, and key confounding factors, such as dietary diversity, physical activity, and parental influence, were not thoroughly studied, potentially affecting result interpretation.

## Conclusion

This study found that almost half of secondary school adolescents in the study area consume fast food regularly, highlighting concern of significant public health. The main significant findings were demographic, economic, and environmental factors such as sex, age, living with other individuals other than parents, food security, disposable income, and home availability of fast foods in prompting dietary behaviors. Given the health risks associated with fast food consumption, targeted interventions are urgently required. These should include school-based health education on healthy consuming, parental engagement in influencing adolescent dietary practices, and strategies that regulate the accessibility and marketing of fast food around educational institutions. Addressing food insecurity and promoting affordable, nutritious food options must be prioritized in public health efforts in the Somali Region and other similar settings.

## Supporting information

**S1 File. Data used in analysis for this study.**
(XLSX)

**S2 File. ANNEX.** Participants information sheet and study questionnaire.
(DOCX)

**S3 File. Inclusivity-in-global-research-questionnaire.**
(DOCX)

## Acknowledgments

The authors express their gratitude to everybody who contributed to this original article at any step.

## Author contributions

**Conceptualization:** Ahmed Mohammed Ibrahim, Nasra Ali Hussein, Mohyadin Abdullahi Ahmed, Girma Tadesse Wadajo, Mohamed Omar Osman, Ramadan Budul Yusuf, Abdilahi Ibrahim Muse, Seid Muhumed Abdilaahi, Raymond Place, Jakob Zinsstag.

**Data curation:** Ahmed Mohammed Ibrahim, Nasra Ali Hussein, Mohyadin Abdullahi Ahmed, Girma Tadesse Wadajo, Mohamed Omar Osman, Ramadan Budul Yusuf, Abdilahi Ibrahim Muse, Seid Muhumed Abdilaahi, Raymond Place, Jakob Zinsstag.

**Formal analysis:** Ahmed Mohammed Ibrahim, Nasra Ali Hussein, Mohyadin Abdullahi Ahmed, Girma Tadesse Wadajo, Mohamed Omar Osman, Ramadan Budul Yusuf, Abdilahi Ibrahim Muse, Seid Muhumed Abdilaahi, Raymond Place, Jakob Zinsstag.

**Investigation:** Ahmed Mohammed Ibrahim, Nasra Ali Hussein, Mohyadin Abdullahi Ahmed, Girma Tadesse Wadajo, Mohamed Omar Osman, Ramadan Budul Yusuf, Abdilahi Ibrahim Muse, Seid Muhumed Abdilaahi, Raymond Place, Jakob Zinsstag.

**Methodology:** Ahmed Mohammed Ibrahim, Nasra Ali Hussein, Mohyadin Abdullahi Ahmed, Girma Tadesse Wadajo, Mohamed Omar Osman, Ramadan Budul Yusuf, Abdilahi Ibrahim Muse, Seid Muhumed Abdilaahi, Raymond Place, Jakob Zinsstag.

**Project administration:** Ahmed Mohammed Ibrahim, Nasra Ali Hussein, Mohyadin Abdullahi Ahmed, Girma Tadesse Wadajo, Mohamed Omar Osman, Ramadan Budul Yusuf, Abdilahi Ibrahim Muse, Seid Muhumed Abdilaahi.

**Resources:** Mohyadin Abdullahi Ahmed, Girma Tadesse Wadajo, Mohamed Omar Osman, Ramadan Budul Yusuf, Abdilahi Ibrahim Muse, Seid Muhumed Abdilaahi.

**Software:** Ahmed Mohammed Ibrahim, Nasra Ali Hussein, Mohyadin Abdullahi Ahmed, Girma Tadesse Wadajo, Mohamed Omar Osman, Ramadan Budul Yusuf, Abdilahi Ibrahim Muse, Seid Muhumed Abdilaahi, Raymond Place, Jakob Zinsstag.

**Supervision:** Ahmed Mohammed Ibrahim, Nasra Ali Hussein, Mohyadin Abdullahi Ahmed, Girma Tadesse Wadajo, Mohamed Omar Osman, Ramadan Budul Yusuf, Abdilahi Ibrahim Muse, Seid Muhumed Abdilaahi, Raymond Place, Jakob Zinsstag.

**Validation:** Ahmed Mohammed Ibrahim, Nasra Ali Hussein, Mohyadin Abdullahi Ahmed, Girma Tadesse Wadajo, Mohamed Omar Osman, Ramadan Budul Yusuf, Abdilahi Ibrahim Muse, Seid Muhumed Abdilaahi, Raymond Place, Jakob Zinsstag.

**Visualization:** Ahmed Mohammed Ibrahim, Nasra Ali Hussein, Mohyadin Abdullahi Ahmed, Girma Tadesse Wadajo, Mohamed Omar Osman, Ramadan Budul Yusuf, Abdilahi Ibrahim Muse, Seid Muhumed Abdilaahi, Raymond Place, Jakob Zinsstag.

**Writing – original draft:** Ahmed Mohammed Ibrahim, Nasra Ali Hussein, Mohyadin Abdullahi Ahmed, Girma Tadesse Wadajo, Mohamed Omar Osman, Ramadan Budul Yusuf, Abdilahi Ibrahim Muse, Seid Muhumed Abdilaahi, Raymond Place, Jakob Zinsstag.

**Writing – review & editing:** Ahmed Mohammed Ibrahim, Nasra Ali Hussein, Mohyadin Abdullahi Ahmed, Girma Tadesse Wadajo, Mohamed Omar Osman, Ramadan Budul Yusuf, Abdilahi Ibrahim Muse, Seid Muhumed Abdilaahi, Raymond Place, Jakob Zinsstag.

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
