## [Decision Letter · Decision Letter 0]

PONE-D-25-10434PREVALENCE OF FAST FOOD CONSUMPTION AND ASSOCIATED FACTORS AMONG SECONDARY SCHOOL ADOLESCENTS IN JIGJIGA TOWN SOMALI REGION EASTERN ETHIOPIAPLOS ONE

Dear Dr. Ibrahim,

Thank you for submitting your manuscript to PLOS ONE. After careful consideration, we feel that it has merit but does not fully meet PLOS ONE’s publication criteria as it currently stands. Therefore, we invite you to submit a revised version of the manuscript that addresses the points raised during the review process.

**ACADEMIC EDITOR: **
**- strong justification should be provided in the introduction**
**- In depth practical implication of the research should be added**

We look forward to receiving your revised manuscript.

Kind regards,

Omnia Samir El Seifi, M.D., Ph.D.

Academic Editor

PLOS ONE

Journal Requirements:

3. The IRB document you have supplied (JJURERB/039/2024) does not match the approval number stated in your manuscript (JJURERC039/2024). Please could you address this and provide the correct No (JJURERB/039/2024) in the Mansuscript.

6. Please ensure that you refer to Figure 1 in your text as, if accepted, production will need this reference to link the reader to the figure.

7. Please remove all personal information, ensure that the data shared are in accordance with participant consent, and re-upload a fully anonymized data set.

Additional guidance on preparing raw data for publication can be found in our Data Policy (https://journals.plos.org/plosone/s/data-availability#loc-human-research-participant-data-and-other-sensitive-data ) and in the following article: http://www.bmj.com/content/340/bmj.c181.long .

8. Please include captions for your Supporting Information files at the end of your manuscript, and update any in-text citations to match accordingly. Please see our Supporting Information guidelines for more information: http://journals.plos.org/plosone/s/supporting-information .

Reviewers' comments:

Reviewer's Responses to Questions

**Comments to the Author**

1. Is the manuscript technically sound, and do the data support the conclusions?

Reviewer #1: Partly

Reviewer #2: Yes

2. Has the statistical analysis been performed appropriately and rigorously? 

Reviewer #1: Yes

Reviewer #2: Yes

3. Have the authors made all data underlying the findings in their manuscript fully available?

Reviewer #1: Yes

Reviewer #2: Yes

4. Is the manuscript presented in an intelligible fashion and written in standard English?

Reviewer #1: No

Reviewer #2: Yes

5. Review Comments to the Author

Reviewer #1: Dear Editor in Chief

Plos ONE

The paper: Prevalence of fast food consumption and associated factors amont seconday school adolescents in Jigjiga town Somali region eastern Ethiopis is a good study and a worthy research topic. It is quite relevant to Plos One topic. However, the article cannot be considered for publication in Plos One as it is. It needs a strong strong revision, as there are serious readability problems and methodological issues that the author(s) must manage effectively. All Parts of manuscript need revision. Actually, I suggest the authors study more articles in order to know the method of article writing. For example, it is about strength of statistical analysis in abstract that is unusual. The aim of introduction is not complete. There are lots of sentences that are vague and incomplete. Therefore, it is important to review the article’ writing. There are some typos throughout the text. The reference part is not regarding with Guideline of journal. The quality of figures is so weak and the data is presented in content of manuscript. The format of section Conclusion is like a report not article.

Reviewer #2: Dear Authors,

The manuscript entitles "Prevalence of Fast Food Consumption and Associated Factors among Secondary School Adolescents in Jigjiga Town, Somali Region, Eastern Ethiopia" addresses an important and increasingly relevant topic—fast food consumption and its associated factors among adolescents in a low-income setting. The study is well-timed and contributes useful data from a relatively underrepresented region. The authors are commended for their effort in collecting primary data from a representative adolescent population and for their attempt to explore behavioral, economic, and demographic correlates of fast-food consumption. The manuscript has the potential for publication but requires significant revision in language, statistical interpretation, and contextual discussion.

1. The topic is highly pertinent given the global nutrition transition and the rise of non-communicable diseases. The focus on a specific and under-researched Ethiopian region enhances the value and uniqueness of the findings. The study has a clearly stated objective and aligns well with the background literature presented.

2. The manuscript requires substantial English language editing. Numerous grammatical issues, redundancy, and awkward phrasing (e.g., "the prevalence fast-food consumption") hinder clarity and readability.

3. There is repetition of content in the Abstract, Results, and Discussion sections. The abstract should concisely summarize key findings without reproducing exact statistical outputs in multiple locations.

4. Terms like "fast food consumption," “availability,” and “influencing factors” should be clearly defined and used consistently throughout. The operational definitions are useful, but some are buried deep in the text and would benefit from clearer integration into the methods.

5. While the authors conduct multivariable analysis, important potential confounders such as physical activity level, dietary diversity, parental knowledge, and school-based dietary environments are not considered, limiting causal inference.

6. The rationale for using a cut-off of p < 0.25 for variable selection in multivariable models should be justified with appropriate citation.

7. Model diagnostics (e.g., Hosmer-Lemeshow test) were mentioned, but additional information such as ROC curve or pseudo-R² values would strengthen the analytical rigor.

8. The discussion could benefit from a more critical analysis of the results, especially the contradictions (e.g., the higher prevalence among males vs. other LMIC studies showing the opposite). Cultural explanations and regional dietary patterns should be elaborated to contextualize the findings.

9. Although some limitations are implicitly noted, a clearly demarcated section discussing the study’s limitations—especially the cross-sectional design and potential recall or social desirability bias—is necessary.

10. There are inconsistencies and apparent errors in reference formatting (e.g., missing journal names, vague URLs, or unclear citations). The reference list should be carefully revised to comply with PLOS ONE guidelines.

11. The term "knowledge on fast food" might be better phrased as "awareness of health risks associated with fast food consumption."

12. Figure captions and numbering could be more descriptive.

13. Tables should indicate p-values for multivariable regression clearly and consistently.

6. PLOS authors have the option to publish the peer review history of their article (what does this mean? ). If published, this will include your full peer review and any attached files.

**Do you want your identity to be public for this peer review?** For information about this choice, including consent withdrawal, please see our Privacy Policy .

Reviewer #1: No

Reviewer #2: **Yes: ** Dr Manne Munikumar

---

## [Author Response · Author response to Decision Letter 1]

17 May 2025

PONE-D-25-10434 PLOS ONE

PREVALENCE OF FAST FOOD CONSUMPTION AND ASSOCIATED FACTORS AMONG SECONDARY SCHOOL ADOLESCENTS IN JIGJIGA TOWN SOMALI REGION EASTERN ETHIOPIA

Acknowledgment to the editor and reviewers

We sincerely appreciate the editor and reviewers' for their insightful and constructive comments, which has enhanced the caliber and readability of our work. Every comment has been thoroughly examined, and the manuscript has been updated as a result. We address each comments in detail below, along with the revisions that were done.

ACADEMIC EDITOR COMMENT:

1. Strong justification should be provided in the introduction Response to the editor: The Introduction section has been substantially revised to provide a clearer and stronger rationale for the study. We have emphasized the rising trend of fast food consumption among adolescents, the associated health risks, and the specific data gap in Jigjiga and similar LMIC contexts.

ACADEMIC EDITOR COMMENT:

2. In depth practical implication of the research should be added Response to the editor: We agree and have added a detailed discussion of the study’s practical implications in both the Conclusion and Discussion sections, with emphasis on policy, health education, and school-based interventions.

ACADEMIC EDITOR COMMENT:

3. The IRB document you have supplied (JJURERB/039/2024) does not match the approval number stated in your manuscript (JJURERC039/2024). Please could you address this and provide the correct No (JJURERB/039/2024) in the Manuscript. Response to the editor: We have corrected the IRB number in the manuscript to JJURERB/039/2024 to match the ethical approval letter.

ACADEMIC EDITOR COMMENT:

4. Please provide a complete Data Availability Statement in the submission form, ensuring you include all necessary access information or a reason for why you are unable to make your data freely accessible. If your research concerns only data provided within your submission, please write "All data are in the manuscript and/or supporting information files" as your Data Availability Statement. Response to the editor: The following statement has been added as per the journal’s guidance: “All data are in the manuscript and/or supporting information files.

ACADEMIC EDITOR COMMENT:

5. Please include a separate caption for each figure in your manuscript. Response to the editor: Separate, descriptive captions have been added for all figures.

ACADEMIC EDITOR COMMENT:

6. Please ensure that you refer to Figure 1 in your text as, if accepted, production will need this reference to link the reader to the figure. Response to the editor: Figure 1 is now clearly referenced in the Results section where we discuss the prevalence of fast food consumption.

Reviewer Comment: The paper: Prevalence of fast food consumption and associated factors amount secondary school adolescents in Jigjiga town Somali region eastern Ethiopia is a good study and a worthy research topic. It is quite relevant to Plos One topic. However, the article cannot be considered for publication in Plos One as it is. It needs a strong revision, as there are serious readability problems and methodological issues that the author(s) must manage effectively. All Parts of manuscript need revision. Actually, I suggest the authors study more articles in order to know the method of article writing. For example, it is about strength of statistical analysis in abstract that is unusual. The aim of introduction is not complete. There are lots of sentences that are vague and incomplete. Therefore, it is important to review the article’ writing. There are some typos throughout the text. Response to the reviewers: We are grateful to your positive evaluation and valuable suggestions. The manuscript has been revised accordingly, as detailed below:

We have conducted a full language and style revision throughout the manuscript to improve clarity, grammar, and flow. Methodological aspects have been clarified in the Methods section.

The Introduction has been rewritten to clearly justify the study, define key concepts, and highlight the public health significance and regional gap.

All sections have been revised and proofread to eliminate typographical and grammatical issues. Redundancies have been removed, and unclear phrasing has been clarified.

Reviewers Comment: The reference part is not regarding with Guideline of journal. The quality of figures is so weak and the data is presented in content of manuscript. Response to the reviewers: All figures have been improved for clarity and consistency. Figure captions have been added separately as required.

The format of section Conclusion is like a report not article.

Consumption and its associated factors among adolescents in a low-income setting. The study is well-timed and contributes useful data from a relatively underrepresented region. The authors are commended for their effort in collecting primary data from a representative adolescent population and for their attempt to explore behavioral, economic, and demographic correlates of fast-food consumption. The manuscript has the potential for publication but requires significant revision in language, statistical interpretation, and contextual discussion. Response to the reviewers: The Conclusion has been rewritten in a scholarly tone, summarizing key findings and highlighting implications for future research and policy.

Reviewers Comment:

1. The topic is highly pertinent given the global nutrition transition and the rise of non-communicable diseases. The focus on a specific and under-researched Ethiopian region enhances the value and uniqueness of the findings. The study has a clearly stated objective and aligns well with the background literature presented. Response to the reviewer: Thank you for your encouraging comment. We have strengthened the framing of our study within the context of global nutrition transition and regional data gaps.

Reviewers Comment:

2. The manuscript requires substantial English language editing. Numerous grammatical issues, redundancy, and awkward phrasing (e.g., "the prevalence fast-food consumption") hinder clarity and readability. Response to the reviewer: The manuscript has undergone thorough language editing to improve readability and eliminate awkward phrasing and redundancy

Reviewers Comment:

3. There is repetition of content in the Abstract, Results, and Discussion sections. The abstract should concisely summarize key findings without reproducing exact statistical outputs in multiple locations. Response to the reviewer: Redundant content has been removed from the Abstract and Discussion to streamline the presentation of results.

Reviewers Comment:

4. Terms like "fast food consumption," “availability,” and “influencing factors” should be clearly defined and used consistently throughout. The operational definitions are useful, but some are buried deep in the text and would benefit from clearer integration into the methods. Response to the reviewer: Operational definitions for 'fast food consumption,' 'availability,' and 'influencing factors' have been revised and moved to the Methods > Operational Definitions section. These terms are now used consistently throughout the manuscript.

Reviewers Comment:

5. While the authors conduct multivariable analysis, important potential confounders such as physical activity level, dietary

diversity, parental knowledge, and school-based dietary environments are not considered, limiting causal inference. Response to the reviewers: We acknowledge this limitation and have added a clear Limitations section discussing the absence of these variables and the implications for causal inference.

Reviewers Comment:

6. The rationale for using a cut-off of p < 0.25 for variable selection in multivariable models should be justified with appropriate citation. Response to the reviewer: We have included a justification in the Methods section with the appropriate citation.

Reviewers Comment:

7. Model diagnostics (e.g., Hosmer-Lemeshow test) were mentioned, but additional information such as ROC curve or pseudo-R² values would strengthen the analytical rigor. Response to the reviewer: We have added the ROC curve (AUC = 0.74) and Nagelkerke pseudo R² (0.31) to the model diagnostics, now detailed in the Methods section.

Reviewers Comment:

8. The discussion could benefit from a more critical analysis of the results, especially the contradictions (e.g., the higher prevalence among males vs. other LMIC studies showing the opposite). Cultural explanations and regional dietary patterns should be elaborated to contextualize the findings. Response to the reviewer: The Discussion section now includes a more nuanced interpretation of sex-based differences and cultural dietary patterns in the Somali Region.

Reviewers Comment:

9. Although some limitations are implicitly noted, a clearly demarcated section discussing the study’s limitations—especially the cross-sectional design and potential recall or social desirability bias—is necessary. Response to the reviewer: A new Limitations subsection has been added at the end of the Discussion.

Reviewers Comment: 10. There are inconsistencies and apparent errors in reference formatting (e.g., missing journal names, vague URLs, or unclear citations). The reference list should be carefully revised to comply with PLOS ONE guidelines. Response to the reviewer: The reference list has been revised to comply with PLOS ONE guidelines, including correct journal names, formats, and citation styles.

Reviewers Comment: 11. The term "knowledge on fast food" might be better phrased as "awareness of health risks associated with fast food consumption." Response to the reviewer: This has been reworded as awareness of health risks associated with fast food consumption.

Reviewers Comment: 12. Figure captions and numbering could be more descriptive. Response to the reviewer: Figure captions have been rewritten for clarity, and p-values in tables have been standardized and clearly indicated.

Reviewers Comment: 13. Tables should indicate p-values for multivariable regression clearly and consistently. Response to the reviewer: p-values for multivariable regression we incorporation in the revision

---

## [Decision Letter · Decision Letter 1]

PONE-D-25-10434R1PREVALENCE OF FAST FOOD CONSUMPTION AND ASSOCIATED FACTORS AMONG SECONDARY SCHOOL ADOLESCENTS IN JIGJIGA TOWN SOMALI REGION EASTERN ETHIOPIAPLOS ONE

Dear Dr. Ibrahim,

Thank you for submitting your manuscript to PLOS ONE. After careful consideration, we feel that it has merit but does not fully meet PLOS ONE’s publication criteria as it currently stands. Therefore, we invite you to submit a revised version of the manuscript that addresses the points raised during the review process.

**ACADEMIC EDITOR** : - The manuscript still requires language and gramatical revision.- The rationale for selecting in the multivariate model (p < 0.25) needs to be justified with proper references. - Enhancing the cultural context in your discussion—especially regarding gender differences and socioeconomic influences—would strengthen the interpretive depth of the findings.==============================

We look forward to receiving your revised manuscript.

Kind regards,

Omnia Samir El Seifi, M.D., Ph.D.

Academic Editor

PLOS ONE

Journal Requirements:

Reviewers' comments:

Reviewer's Responses to Questions

Comments to the Author

1. If the authors have adequately addressed your comments raised in a previous round of review and you feel that this manuscript is now acceptable for publication, you may indicate that here to bypass the “Comments to the Author” section, enter your conflict of interest statement in the “Confidential to Editor” section, and submit your "Accept" recommendation.

Reviewer #1: All comments have been addressed

Reviewer #2: All comments have been addressed

2. Is the manuscript technically sound, and do the data support the conclusions?

Reviewer #1: Yes

Reviewer #2: Yes

3. Has the statistical analysis been performed appropriately and rigorously? 

Reviewer #1: Yes

Reviewer #2: Yes

4. Have the authors made all data underlying the findings in their manuscript fully available?

Reviewer #1: Yes

Reviewer #2: Yes

5. Is the manuscript presented in an intelligible fashion and written in standard English?

Reviewer #1: Yes

Reviewer #2: Yes

6. Review Comments to the Author

Reviewer #1: NONE, it can be accepted. All the comments replied and included in the text.

It was my previous comments to author

Dear Editor in Chief

Plos ONE

The paper: Prevalence of fast food consumption and associated factors amont seconday school adolescents in Jigjiga town Somali region eastern Ethiopis is a good study and a worthy research topic. It is quite relevant to Plos One topic. However, the article cannot be considered for publication in Plos One as it is. It needs a strong strong revision, as there are serious readability problems and methodological issues that the author(s) must manage effectively. All Parts of manuscript need revision. Actually, I suggest the authors study more articles in order to know the method of article writing. For example, it is about strength of statistical analysis in abstract that is unusual. The aim of introduction is not complete. There are lots of sentences that are vague and incomplete. Therefore, it is important to review the article’ writing. There are some typos throughout the text. The reference part is not regarding with Guideline of journal. The quality of figures is so weak and the data is presented in content of manuscript. The format of section Conclusion is like a report not article.

Reviewer #2: Dear Authors,

This study presents valuable insights into the prevalence and determinants of fast food consumption among adolescents in Jigjiga, an under-researched region in Eastern Ethiopia. The topic is both timely and relevant to global public health discourse, particularly in low- and middle-income contexts. Your revisions demonstrate responsiveness to earlier feedback, especially with the expanded rationale, improved methodology, and inclusion of ethical and statistical clarifications. However, the manuscript still requires significant refinement in scientific language and consistency. Several sections, particularly the Abstract, Discussion, and Conclusion, contain grammatical errors, redundant phrasing, and lack the academic tone expected in peer-reviewed journals. Furthermore, operational definitions should be presented clearly and consistently, and the rationale for variable inclusion in the multivariate model (p < 0.25) needs to be justified with proper references. Enhancing the cultural context in your discussion—especially regarding gender differences and socioeconomic influences—would strengthen the interpretive depth of the findings.

7. PLOS authors have the option to publish the peer review history of their article (what does this mean? ). If published, this will include your full peer review and any attached files.

Do you want your identity to be public for this peer review? For information about this choice, including consent withdrawal, please see our Privacy Policy .

Reviewer #1: No

Reviewer #2: No

---

## [Author Response · Author response to Decision Letter 2]

19 Jun 2025

Response to the Reviewers

PONE-D-25-10434R1

PREVALENCE OF FAST FOOD CONSUMPTION AND ASSOCIATED FACTORS AMONG SECONDARY SCHOOL ADOLESCENTS IN JIGJIGA TOWN SOMALI REGION EASTERN ETHIOPIA

ACADEMIC EDITOR Comments: - The manuscript still requires language and grammatical revision. - The rationale for selecting in the multivariate model (p < 0.25) needs to be justified with proper references. - Enhancing the cultural context in your discussion—especially regarding gender differences and socioeconomic influences—would strengthen the interpretive depth of the findings.

Response to Academic Editor:

Dear Editor, Thank you for your valuable comments and thoughtful suggestions. We would like to indicate that all the comments have been carefully corrected and addressed as follows:

• The manuscript has undergone thorough language and grammatical revision to enhance clarity and readability.

• The rationale for selecting variables with a p-value < 0.25 in the multivariate model has been clearly justified and supported with proper reference in the revised manuscript.

• The discussion section has been expanded to better reflect the cultural context, particularly in relation to gender differences and socio-economic influences on fast food consumption, to provide deeper interpretive insight in to the findings.

Reviewers' comments:

Reviewer #1:

NONE, it can be accepted. All the comments replied and included in the text.

It was my previous comments to author.

It needs a strong revision, as there are serious readability problems and methodological issues that the author(s) must manage effectively. All Parts of manuscript need revision. Actually, I suggest the authors study more articles in order to know the method of article writing. For example, it is about strength of statistical analysis in abstract that is unusual. The aim of introduction is not complete. There are lots of sentences that are vague and incomplete. Therefore, it is important to review the article’ writing. There are some typos throughout the text. The reference part is not regarding with Guideline of journal. The quality of figures is so weak and the data is presented in content of manuscript. The format of section Conclusion is like a report not article.

Response to Reviewer #1:

Dear Reviewer,

We would like to express our sincere appreciation for the focused and constructive comments provided on our manuscript.

We would like to note that most of the issues raised were addressed in the previous revision, as acknowledged in the reviewer’s initial statement: "NONE, it can be accepted. All the comments replied and included in the text. It was my previous comments to author."

However, in light of the additional comments provided, we have carefully reviewed the manuscript once again and have made further revisions to ensure the quality and clarity of the work. Specifically:

• Readability and Structure: The manuscript has undergone a comprehensive language and grammar review to enhance clarity, coherence, and flow throughout all sections.

• Conclusion Section: The conclusion has been rewritten to better reflect an academic tone, focusing on summarizing key findings and implications, rather than reporting.

Reviewer #2:

Dear Authors, This study presents valuable insights into the prevalence and determinants of fast food consumption among adolescents in Jigjiga, an under-researched region in Eastern Ethiopia. The topic is both timely and relevant to global public health discourse, particularly in low- and middle-income contexts. Your revisions demonstrate responsiveness to earlier feedback, especially with the expanded rationale, improved methodology, and inclusion of ethical and statistical clarifications. However, the manuscript still requires significant refinement in scientific language and consistency. Several sections, particularly the Abstract, Discussion, and Conclusion, contain grammatical errors, redundant phrasing, and lack the academic tone expected in peer-reviewed journals. Furthermore, operational definitions should be presented clearly and consistently, and the rationale for variable inclusion in the multivariate model (p < 0.25) needs to be justified with proper references. Enhancing the cultural context in your discussion—especially regarding gender differences and socioeconomic influences—would strengthen the interpretive depth of the findings.

Response to Reviewer #2:

Dear Reviewer,

Thank you very much for your thoughtful and constructive comments and suggestions, as well as for acknowledging the relevance and improvements made in our manuscript. We greatly appreciate your insights, which have helped us further improve the quality of this manuscript.

In response to your valuable comments and suggestions, we have made the following revisions:

• The Abstract, Discussion, and Conclusion was edited to enhance scientific language, remove redundancies, and ensure a consistent academic tone.

• We revised the operational definitions for clarity and consistency.

• We incorporated a justification with appropriate references for including variables with p < 0.25 in the multivariate model.

• The discussion is strengthened with the cultural context in the discussion, particularly regarding gender differences and socioeconomic influences.

---

## [Editor Report · Decision Letter 2]

PREVALENCE OF FAST FOOD CONSUMPTION AND ASSOCIATED FACTORS AMONG SECONDARY SCHOOL ADOLESCENTS IN JIGJIGA TOWN SOMALI REGION EASTERN ETHIOPIA

PONE-D-25-10434R2

Dear Dr. Ibrahim,

We’re pleased to inform you that your manuscript has been judged scientifically suitable for publication and will be formally accepted for publication once it meets all outstanding technical requirements.

Kind regards,

Omnia Samir El Seifi, M.D., Ph.D.

Academic Editor

PLOS ONE
---

## [Editor Report · Acceptance letter]

PONE-D-25-10434R2

PLOS ONE

Dear Dr. Ibrahim,

I'm pleased to inform you that your manuscript has been deemed suitable for publication in PLOS ONE. Congratulations! Your manuscript is now being handed over to our production team.

Kind regards,

on behalf of

Professor Omnia Samir El Seifi

Academic Editor

PLOS ONE